# Best Practices for Body Temperature Measurement with Infrared Thermography: External Factors Affecting Accuracy

**DOI:** 10.3390/s23188011

**Published:** 2023-09-21

**Authors:** Siavash Mazdeyasna, Pejman Ghassemi, Quanzeng Wang

**Affiliations:** Center for Devices and Radiological Health, U.S. Food and Drug Administration, Silver Spring, MD 20993, USA; siavash.mazdeyasna@fda.hhs.gov (S.M.); pejman.ghassemi@fda.hhs.gov (P.G.)

**Keywords:** elevated body temperature, infrared thermograph, thermography, ISO/TR 13154, viewing angle, external temperature reference source, ambient temperature, relative humidity, atmosphere transmittance, environmental effects, accuracy

## Abstract

Infrared thermographs (IRTs) are commonly used during disease pandemics to screen individuals with elevated body temperature (EBT). To address the limited research on external factors affecting IRT accuracy, we conducted benchtop measurements and computer simulations with two IRTs, with or without an external temperature reference source (ETRS) for temperature compensation. The combination of an IRT and an ETRS forms a screening thermograph (ST). We investigated the effects of viewing angle (*θ*, 0–75°), ETRS set temperature ($T_{ETRS}$, 30–40 °C), ambient temperature ($T_{atm}$, 18–32 °C), relative humidity (RH, 15–80%), and working distance (*d*, 0.4–2.8 m). We discovered that STs exhibited higher accuracy compared to IRTs alone. Across the tested ranges of $T_{atm}$ and RH, both IRTs exhibited absolute measurement errors of less than 0.97 °C, while both STs maintained absolute measurement errors of less than 0.12 °C. The optimal $T_{ETRS}$ for EBT detection was 36–37 °C. When *θ* was below 30°, the two STs underestimated calibration source (CS) temperature ($T_{CS}$) of less than 0.05 °C. The computer simulations showed absolute temperature differences of up to 0.28 °C and 0.04 °C between estimated and theoretical temperatures for IRTs and STs, respectively, considering *d* of 0.2–3.0 m, $T_{atm}$ of 15–35 °C, and RH of 5–95%. The results highlight the importance of precise calibration and environmental control for reliable temperature readings and suggest proper ranges for these factors, aiming to enhance current standard documents and best practice guidelines. These insights enhance our understanding of IRT performance and their sensitivity to various factors, thereby facilitating the development of best practices for accurate EBT measurement.

## 1. Introduction

Elevated body temperature (EBT) has emerged as a critical indicator for various infectious diseases that have triggered outbreaks, epidemics, and pandemics in recent years. Notable instances include the severe acute respiratory syndrome (SARS) epidemic in 2003, the influenza A (H1N1) epidemic in 2009, the middle east respiratory syndrome (MERS) outbreak in 2012, the Ebola virus disease (EVD) epidemic in 2014, the coronavirus (COVID-19) pandemic from 2020 to 2022, and the Mpox virus (MPV) outbreak in 2022 [1,2,3,4,5,6,7,8,9]. Although relying solely on EBT screening may not suffice as an isolated preventive measure, it can serve as a valuable component within a comprehensive risk management strategy, particularly during the early stages of an epidemic when vaccines and testing resources may be limited or unavailable.

Thermal modalities, including infrared thermographs (IRTs, also known as thermal/infrared cameras/imagers) and non-contact infrared thermometers (NCITs), offer safe, non-contact, and non-invasive methods for EBT screening. These modalities have gained popularity as screening tools for identifying individuals with EBT in public areas during infectious disease pandemics, as they enable fast and real-time detection of infrared (IR) energy radiated by a person’s face. NCITs measure temperature at a single point using a sensor with one or a few pixels. On the other hand, IRTs provide a temperature map of a large area using a sensor with thousands of pixels, which allows for greater flexibility in assessing different regions of the face and improves the effectiveness of temperature estimation [10,11]. Recent studies and evidence in the literature suggest that qualified IRTs, when used properly, can offer superior accuracy in estimating EBT compared to NCITs [12,13,14]. The ability of IRTs to capture temperature distributions across a large area can contribute to more accurate readings and mitigates the impact of localized temperature variations that may occur with NCITs.

In recent years, IRT technology has undergone significant advancements and demonstrates potential across a spectrum of applications [15]. These applications include cancer detection [16], sports medicine [17], foot thermoregulation study [18], sinusitis detection [19], arthritis diagnosis [20], neonatal disease detection [21,22], multiple sclerosis evaluation [23], evaluation of diabetes-associated vascular disorders [24], dermatology [25], pain monitoring [26], dentistry [27], surgical procedures [28], etc. In this study, our focus is on the use of IRTs for EBT measurements.

Many parameters can affect the temperature measurement accuracy of an IRT. Moreira et al. presented a comprehensive checklist of parameters that might impact the measurement of skin temperature using IRTs in sports and exercise medicine, including individual-specific data (e.g., age, sex, body mass index, height, ethnicity), environmental factors (e.g., temperature, humidity, air flow, and sources of infrared radiation), contextual details (e.g., skin dryness, recent activity, measurement time), and instrumental variables (e.g., camera model, software, emissivity configuration) [17]. Zagrodny proposed best practices for accurate temperature measurements with IRTs, including subject preparation, laboratory conditions, and imaging practices [29]. Similar best practices for proper deployment, implementation, and operational practices are also recommended by international standard documents [30] and guidelines [31]. It is challenging to encompass all the factors that can influence IRT accuracy. Therefore, in this study, our focus is solely on environmental parameters (ambient temperature and relative humidity (RH)) and deployment parameters (working distance, viewing angle, and the set temperature of the external temperature reference source (ETRS)).

Our previous work has shown that if specific standardized requirements for basic safety and essential performance of IRTs are met and proper deployment, implementation, and operational guidelines are followed, IRTs can be used to accurately detect EBT [10]. The International Electrotechnical Commission (IEC) standard IEC 80601-2-59:2017 [32] recommends requirements for basic safety (e.g., electrical/mechanical/radiation hazards) and laboratory accuracy of IRTs for detecting EBT. While the basic safety requirements are straightforward, some laboratory accuracy test methods recommended by this standard need to be optimized. We previously implemented and evaluated these methods and proposed modifications to improve their effectiveness [33]. An FDA Regulatory Science Tool to evaluate IRT laboratory performance has also been developed [34]. However, no standards have been established to define requirements for IRT clinical accuracy. We previously conducted a clinical study involving over 1000 subjects to evaluate and compare different metrics and thresholds for quantifying IRT clinical accuracy [14,35].

While previous studies have addressed factors influencing IRT accuracy and recommended optimal temperature measurement practices, a gap remains in terms of systematic, quantitative analyses. Many studies reported their research conditions without quantitatively evaluating their impact on IRT accuracy. Other sources, including certain international standard documents [30], proposed ideal conditions and best practices for IRT use. However, consistently achieving these conditions can be impractical, as exemplified during the COVID-19 pandemic when temperature assessments frequently occurred at suboptimal building entrances. Amidst these considerations, a critical question arises: To what extent can deviations from ideal circumstances be tolerated while maintaining an acceptable level of temperature measurement accuracy? This central query forms the primary focus of our research. Specifically, we concentrate on various environmental and deployment factors within this context. Through this investigation, our aim is to offer practical insights into the permissible range of departure from ideal conditions for accurate temperature measurements. Our study examines environmental conditions including ambient temperature and RH, and deployment parameters including working distance, viewing angle, and the ETRS set temperature. We have chosen to scrutinize these parameters because they are inadequately defined in the international standard document [30] or defined within a narrow range that is unattainable in certain practical applications.

An ETRS—a blackbody radiator with known emissivity and temperature—is often essential to body temperature measurement with an IRT. Our previous study [33] has demonstrated that utilizing an ETRS for offset temperature compensation can improve the accuracy of an IRT when the stability and drift of the IRT alone cannot satisfy the standard accuracy requirements [32]. When combined, an IRT and an ETRS create a screening thermograph (ST) [32,33]. The ETRS temperature should ideally be set near a chosen diagnostic threshold temperature at the measurement site, which is the temperature above which a fever is considered to be present (note: Some IRTs and thermometers measure temperature at one site and convert it to a temperature at a reference site [14]). The ISO/TR 13154 standard document provides an example of setting the ETRS temperature at 35 °C [30]. However, there is currently no consensus on the optimal ETRS set temperature for IRT applications, and the impact of the ETRS set temperature on IRT accuracy has not been sufficiently investigated.

Viewing angle is a significant factor affecting IRT accuracy. The IEC 80601-2-59 and ISO/TR 13154 standard documents [30,32] specify that the target plane should be perpendicular to the IRT optical axis. Research has demonstrated that the emissivity of a flat object surface decreases as the viewing angle increases [36,37]. Consequently, if the IRT emissivity setting is fixed, the IRT will underestimate the object’s temperature at large angles. Several studies have explored concerns related to viewing angles in different applications [38,39,40]. However, these studies were focused on non-medical applications, such as infrastructure, building, and satellite imaging, especially at long working distances or short/medium wavelengths. Hori et al. developed a model for angular-dependent emissivity, but specifically for snow and ice in the 8–13 μm infrared band [41]. Cheng et al. investigated the impact of viewing angle on IRT accuracy, but their study was limited to medium/short wavelength (3–5 µm) IRTs [36]. Therefore, there is a need for a study that simulates a medical application scenario and examines the effect of viewing angle on IRT accuracy in that context.

Accurate body temperature measurement using IRTs and NCITs requires controlled ambient/atmospheric temperature, RH, and distance ranges. The ISO/TR 13154 standard document recommends conducting temperature measurements with IRTs indoors at temperatures of 20 °C to 24 °C and RH between 10% and 50% [30]. The ASTM E1965-98 standard specifies an operating temperature range of 16 °C to 40 °C and an RH range of up to 95% for infrared thermometer applications [42]. Environmental discomfort can affect thermoregulation and acclimatization of the human body [43], potentially impacting the relationship between the temperatures at the measurement site and the reference site [14]. Ambient temperature and RH can also affect skin emissivity by affecting skin moisture levels [44]. Temperature measurement distance is typically determined by the number of pixels required to cover the facial region in thermal images [32]. Additionally, ambient temperature, RH, and measuring distance collectively influence the atmospheric transmittance, which directly affects the accuracy of IRT temperature readings. While the effect of ambient temperature, RH, and working distance on temperature estimation or atmospheric transmittance has been investigated for different NCITs [45] and IRTs [38,46,47,48], there is a lack of consistent data across different documents. Thus, a systematic study is still needed to fully understand the effects of ambient temperature, RH, and measuring distance.

The purpose of this study is to objectively and quantitatively assess the effects of external factors relevant to the validity and practical implementation of IRTs through benchtop measurements and computer simulations. We focused on evaluating the performance of two specific IRTs under varying environmental conditions such as ambient temperature and ambient RH, as well as deployment parameters including working distance, viewing angle, and the ETRS set temperature. We seek to provide valuable insights into the effects of these factors on IRT performance. Through our assessment of these factors across wide ranges, we aim to lay the foundation for appropriate specifications in standard documents, ensuring precise temperature measurements. This contribution aids in advancing, enhancing, deploying, and regulating IRT technologies for highly accurate EBT screening. Preliminary results from this research have been presented at a conference [49], indicating the ongoing progress and dissemination of our findings in this area.

## 2. IRT Theory

Planck’s law [50] describes the spectral radiance of an ideal blackbody source (an object with emissivity equal to one) in thermal equilibrium at a given absolute temperature as:(1)$$L_{e,\Omega ,\lambda }=\frac{2\cdot h\cdot c^{2}}{\lambda^{5}}{\left[e^{\left(\frac{h\cdot c}{k\cdot \lambda \cdot T}\right)}-1\right]}^{-1}$$
where $L_{e,\Omega ,\lambda }$ is the spectral radiance (unit W·m^−2^·sr^−1^·µm^−1^) for a blackbody at a specific wavelength λ (unit µm) and absolute temperature *T* (unit K). The equation incorporates several fundamental constants: h is the Planck constant equal to 6.6261 × 10^−34^ J·s, c is the speed of light equal to 2.9979 × 10^8^ m·s^−1^, and k is the Boltzmann constant equal to 1.3806 × 10^−23^ J·K^−1^ [51].

For a Lambertian radiating surface, the radiance it reflects or emits is equal in all directions (isotropic). To calculate its spectral radiant emittance, denoted as $M_{e,\lambda }$ (in unit of W·m^−2^·µm^−1^), the Nusselt analog method can be employed:(2)$$M_{e,\lambda }=\pi \cdot L_{e,\Omega ,\lambda }$$

Integration of Planck’s law equation for spectral radiant exitance over all wavelengths can obtain the Stefan–Boltzmann formula for a blackbody object as [38,52,53]:(3)$$M_{e}=\int_{0}^{\infty }M_{e,\lambda }d\lambda =\int_{0}^{\infty }\frac{2\pi \cdot h\cdot c^{2}}{\lambda^{5}\cdot \left[e^{\left(\frac{h\cdot c}{k\cdot \lambda \cdot T}\right)}-1\right]}d\lambda =\frac{2\cdot \pi^{5}\cdot k^{4}\;}{15\cdot c^{2}\cdot h^{3}}\cdot T^{4}=\sigma \cdot T^{4}$$
where $M_{e}$ (unit W·m^−2^) is the radiant exitance of the object at temperature *T*, and σ is the Stefan–Boltzmann constant equal to 5.67 × 10^−8^ W·m^−2^·K^−4^. Planck’s law and Stefan–Boltzmann law are derived based on the assumption that the object is in thermal equilibrium.

Most objects in everyday life are called graybodies since their emissivity ε (0 ≤ ε ≤ 1) is less than one. The Stefan–Boltzmann formula for a graybody object can be modified as the following equation, assuming ε is the same across all wavelengths [54,55]:(4)$$M_{e}=\varepsilon \cdot \sigma \cdot T^{4}$$

An object can also reflect thermal radiation originating from other objects (e.g., sun), for a total amount of $\left(1-\varepsilon \right)\cdot \sigma \cdot T_{refl}{}^{4}$ radiant flux per unit area (radiant flux density), where $T_{refl}$ is the reflected temperature—the temperature of the energy incident upon and reflected from the measurement surface of the object [56]. In rare cases, a small amount of incident radiation might pass through the graybody, which is usually ignored. Radiosity is the total radiant flux leaving (emitted, reflected, and transmitted by) an object’s surface per unit area. When an IRT is used to detect the object’s surface temperature, radiation from the object will reach the IRT sensor after passing through the atmosphere, and therefore should be multiplied by the atmosphere transmittance τ (0 ≤ τ ≤ 1). The value of τ can be affected by environmental factors, such as measuring distance, RH, and atmosphere temperature, which will be discussed in detail in Section 3.4. The IRT sensor will also receive a small amount of $\left(1-\tau \right)\cdot \sigma \cdot T_{atm}^{4}$ radiation from the atmosphere with temperature $T_{atm}$. Figure 1 illustrates the total radiant flux received by an IRT. The total radiation ($E_{e,total}$, unit W·m^−2^) received by the IRT sensor can be expressed as follows [55,57]:(5)$$E_{e,total}=\varepsilon \cdot \tau \cdot \sigma \cdot T^{4}+\left(1-\varepsilon \right)\cdot \tau \cdot \sigma \cdot T_{refl}^{4}+\left(1-\tau \right)\cdot \sigma \cdot T_{atm}^{4}$$Based on Equation (5), the object temperature can be calculated as:(6)$$T=\sqrt[{4}]{\frac{E_{e,total}}{\varepsilon \cdot \tau \cdot \sigma }-\frac{\left(1-\varepsilon \right)\cdot T_{refl}^{4}}{\varepsilon }-\frac{\left(1-\tau \right)\cdot T_{atm}^{4}}{\varepsilon \cdot \tau }}$$

In Equation (6), $E_{e,total}$ and $T_{refl}$ [56] can be measured with an IRT sensor, and $T_{atm}$ can be measured with a thermometer. If there is no external source of IR radiation around the IRT, $T_{refl}$ can be considered the same as $T_{atm}$. If we know ε and τ, then *T* can be calculated.

The emissivity of an object’s surface is generally a function of the surface temperature *T*, wavelength λ, and the direction of the emitted radiation (viewing angle) θ, ε=ε(T, λ, θ). We can fix θ to remove this factor and multiply ε=ε(T, λ) to Planck’s equation (Equation (1)), then integrate Planck’s equation to get a specific Stefan–Boltzmann equation for a given object. Based on the ε(T, λ) function, the specific Stefan–Boltzmann equation might be rather complex without an analytical expression. ε may be assumed constant for the case of IRT body temperature measurement because human skin emissivity does not change significantly over the narrow temperature range (e.g., 34–39 °C) and wavelength range (e.g., 7–14 µm) involved [58]. Equations (4)–(6) assume constant ε.

The Stefan–Boltzmann law gives total radiant exitance over all wavelengths from zero to infinity. However, in practical applications, an IRT sensor cannot detect radiation over all wavelengths. For example, many IRT sensors detect radiation in the 7 to 14 µm range. Therefore, a mathematical model to approximate the Stefan–Boltzmann equation for an IRT sensor can be used in the sensor software. While this model might be more than one, we discussed one model we used in Appendix A.

## 3. Methodology

Two long-wavelength (i.e., 7–14 µm) IRTs were used to evaluate the effects of environmental conditions (ambient temperature, RH) and deployment parameters (working distance, viewing angle, and ETRS set temperature) on IRT laboratory accuracy in a controlled lab environment simulating medical applications.

### 3.1. Experimental Setup and Test Method

Bench tests were performed in a closed room with air vents closed. Table 1 lists the devices used in this study with their specifications described in our previous publication [33]. A heater and a humidifier were used to control ambient temperature and RH. Once the desired condition was achieved, they were turned off during the IRT data acquisition to eliminate the effect of airflow. Since our data acquisition was fast (from a few seconds up to 15 min) and the ambient temperature and RH were relatively stable (do not detect change within 30 min), turning off the heater and humidifier did not adversely impact the tests. Figure 2 demonstrates the experimental setups, where the two IRTs were positioned adjacent to each other on a tripod to simultaneously measure the temperatures of both ETRS and calibration source (CS), which are both commercial blackbodies. The ETRS and CS were assembled on a cart with the same height as the IRTs. The ETRS functioned as a reference for IRT offset compensation during temperature measurement, while the CS served as a test target [10,14,33], allowing for the evaluation and comparison of the performance of IRTs. When an IRT and the ETRS work together to provide temperature readings, they form a screening thermograph (ST). The emissivity values of both ETRS and CS are 0.98 ± 0.02, which closely resemble the emissivity of human skin [58]. To comply with the IEC 80601-2-59 standard, the workable target plane (WTP) image was required to have a minimum resolution of 240 × 180 pixels. Unless stated otherwise, both ETRS and CS surfaces were placed within the WTP at a working distance of *d* = 0.8 m to satisfy this minimum resolution requirement of 240 × 180 pixels, as discussed in our previous publication [33].

Since our goal was to evaluate IRT or ST performance for EBT measurement (typical skin surface temperature), the set temperatures of ETRS ($T_{ETRS}$) and CS ($T_{CS}$) were around the threshold temperature range (36–37 °C for skin temperature [10]). The IRTs measured $T_{ETRS}$ and $T_{CS}$ simultaneously. The $T_{CS}$ directly measured by the IRTs without any temperature compensation are referred to as $T_{IRT}$. After undergoing a two-step temperature compensation process, the resulting temperatures are referred to as $T_{ST}$ (previously [10], we called the measured $T_{CS}$ after temperature compensation $T_{IRT}$). The first step of temperature compensation based on the difference between set $T_{ETRS}$ and measured $T_{ETRS}$ has been described in detail in our previous publication [10]. When both CS and ETRS were set to 37 °C and considering the higher accuracy of the CS compared to the ETRS, we employed the difference between the measured $T_{ETRS}$ and measured $T_{CS}$ for the second-step temperature compensation. This approach allowed us to account for any variations between the two measurements and ensure more precise temperature adjustments.

### 3.2. Effect of ETRS Set Temperature

The ETRS should be set close to the threshold temperature (the temperature above which a person would be considered to have a fever) to compensate for any offset and improve the accuracy of temperature measurements [33]. The threshold temperatures for different body sites may vary, and the temperature measured at the measurement site can be converted to a different value at a reference site [14]. To evaluate the effects of the ETRS set temperature ($T_{ETRS}$) on the accuracy of the STs in measuring the CS (i.e., the test target) temperature, we performed experiments where we set the CS temperature ($T_{CS}$) at various values including 30, 34, 35, 36, 37, 38, and 40 °C. Throughout these experiments, we maintained a constant $T_{ETRS}$. Multiple series of experiments were conducted, each with a different $T_{ETRS}$ within the range of 30 °C to 40 °C. In each experiment, a series of thermograms consisting of ten consecutive images were captured at a frame rate of 30 Hz. These thermograms were then averaged to obtain the measured CS temperature using a 50 × 50 pixels area as a region of interest. The measured CS temperature without temperature compensation ($T_{IRT}$) and with temperature compensation ($T_{ST}$) (see Section 3.1 for details) were compared with $T_{CS}$ to calculate the measurement error. The effect of $T_{ETRS}$ on the CS measurement error is discussed in Section 4.1. Our previous study has shown that the threshold temperature based on the whole-face maximum is between 36 °C and 37 °C (Tables 5 and 6 in [10]). The $T_{ETRS}$ was set at 37 °C to investigate the effects of the other external factors, such as ambient temperature, RH, working distance, and viewing angle, on the accuracy of temperature measurement.

### 3.3. Effect of Viewing Angle

The emissivity ε of a surface can be affected by the viewing angle [36,37,41]. Consequently, any deviation from the zero-degree viewing angle (i.e., the angle between the IRT optical axis and surface normal is zero) due to setup misalignment or an object’s surface curvature, has the potential to introduce measurement artifacts. To quantitatively investigate the effect of viewing angle, a series of measurements were conducted to simulate a fever screening scenario. The IRTs were utilized at a relatively short working distance of 1 m in a typical hospital and office environment: the ambient temperature $T_{atm}$ was 23 °C and RH was 25%. The same method described in Section 3.1 was applied, except that the CS (target test) was rotated from 0° to 75°, with a step size of 5°. At each angle, thermal images were captured sequentially using both IRTs, and the $T_{IRT}$ and $T_{ST}$ values were obtained by averaging a 20 × 20 pixel area, except for large viewing angles (15 × 20, 10 × 20, and 5 × 20 pixels were averaged for the viewing angles of 65°, 70°, and 75°, respectively for IRT-1; 15 × 20 and 10 × 20 pixels were averaged for the viewing angles of 70° and 75°, respectively for IRT-2). The experiment was conducted four times, with the CS rotated to the right twice and to the left twice. The average values from these four repetitions were utilized to illustrate the effect of viewing angles on temperature measurement.

### 3.4. Effect of Environmental Factors (Ambient Temperature, RH) and Working Distance

To ensure reliable and consistent temperature measurements, the ISO/TR 13154 standard document recommends maintaining a stable indoor environment with an ambient temperature range of 20 °C to 24 °C and an ambient RH range from 10% to 50% [30]. The IEC 80601-2-59 ED 2017 standard recommends maintaining ambient RH below 50% and ambient temperature below 24 °C [32]. In this paper, we only focus on the effects of ambient temperature and RH on atmospheric transmission. Although an uncomfortable environment can affect IRT accuracy by changing skin moisture and emissivity, thermoregulation and acclimatization of the human body, and the atmosphere transmittance of infrared radiation. 

The atmospheric transmittance τ is typically close to one and can be estimated based on the working distance (i.e., the distance between the object and IRT), atmospheric temperature, and ambient RH as follows [47,48,57]:(7)$$\tau \left(d,\omega \right)=K_{atm}\;\cdot e^{\left[-\sqrt{d}\cdot \left(\alpha_{1}+\beta_{1}\cdot \sqrt{\omega }\right)\right]}+(1-K_{atm})\;\cdot e^{\left[-\sqrt{d}\cdot \left(\alpha_{2}+\beta_{2}\cdot \sqrt{\omega }\right)\right]}$$
where $K_{atm}$ is the scaling factor for the atmospheric damping, d is working distance, $\alpha_{1}$ and $\alpha_{2}$ are attenuation factors for a dry atmosphere without water vapor, $\beta_{1}$ and $\beta_{2}$ are attenuation factors for water vapor, and ω is a coefficient indicating the content of water vapor in the atmosphere, which can be estimated as [47,48,57]:(8)$$\omega \left(\omega_{\%},T_{atm\_C}\right)=\omega_{\%}\cdot \;e^{\left(h_{1}+h_{2}\cdot T_{atm\_C}+h_{3}\cdot T_{atm\_C}^{2}+h_{4}\cdot T_{atm\_C}^{3}\right)}$$
where $\omega_{\%}$ is the ambient RH, $h_{1}$, $h_{2}$, $h_{3}$, and $h_{4}$ are constants, and $T_{atm\_C}$ is the atmospheric temperature in Celsius ($T_{atm}$ is in Kelvin in this paper). The scaling factors, attenuation factors, and other constants in Equations (7) and (8) which determine τ for a thermal signal, are wavelength-dependent and assumed constant in the spectral range of 7–14 µm in our study.

Previous research has reported specific values for these parameters for a long-wavelength IRT (ThermaCAM PM 595, FLIR Systems Inc., Nashua, NH, USA) [59]. The IRT-1 camera and the ThermaCAM PM 595 camera share a similar spectral range of 7.5 µm to 13 µm, which is comparable to the spectral range of 7 µm to 14 µm for the IRT-2 camera. It is important to note that slight variations in nominal sensor specifications may exist between different manufacturers, even for the same sensor chip. Therefore, the same values from Table 2 were utilized to compute τ for both IRT-1 and IRT-2.

In our study, the IRTs are uncooled microbolometers sensitive to long-wavelength IR radiation (the detail of the IRTs can be found in our previous studies [10,14,33,60]). The estimation of τ was based on the characteristics of the specific IRTs used in the study. The value of τ decreases with increasing distance and RH. The FLIR software ResearchIR Max 4 allows the operator to input the working distance, reflected temperature, RH, and ambient/atmospheric temperature to calculate τ based on Equations (7) and (8). The calculated value matched our manually calculated value. Therefore, we used the software-calculated value directly for IRT-1. However, for IRT-2, we manually calculated *τ*.

The ISO/TR 13154 standard document recommends an ambient temperature range of 20 °C to 24 °C for temperature measurements with IRTs [30]. The ASHRAE standard 55 recommends an indoor temperature of 19.4 °C to 27.8 °C for thermal comfort purposes [61]. Additionally, our previous clinical study involving over 1000 subjects was conducted over a broader range of ambient temperature (20 °C to 29 °C), but still showed that qualified IRT systems may offer higher diagnostic efficacy compared to commercial NCITs [14]. Table 3 lists the ambient temperature and RH ranges in different studies. In this study, we selected wider ranges of ambient temperature, RH, and working distance for bench tests (second-to-last row in Table 3). We also conducted computer simulations, as discussed in Section 3.5. Due to resource limitations in terms of available equipment like heater, humidifier, and laboratory space, computer simulations enable the study of ambient temperature, RH, and working distance beyond the ranges achievable in our experimental setup (last row in Table 3).

To explore the effects of environmental factors on temperature measurement, a humidifier and a heater were utilized to control the test environment, replicating various conditions. The experimental setup, which includes the IRTs and blackbodies, remains consistent with the setup described in Section 3.1. The IRTs were utilized at a relatively short working distance of 0.8 m. The temperatures of both ETRS and CS were maintained at 37 °C, which can be justified by the findings discussed in Section 4.1. When the effect of RH was evaluated, the ambient temperature was maintained at 24 °C, while the RH varied in the range of 15% to 80% with a step size of 5%. When the effect of ambient temperature was evaluated, the RH was maintained at 35%, and the ambient temperature was adjusted incrementally from 18 °C to 32 °C with a step size of 1 °C. Similar to our previous publication [33], the average temperature of a 50 × 50 pixels region was calculated to determine the ETRS and CS temperatures.

To investigate the effect of working distance on temperature measurement, the ambient temperature and RH were maintained at 24 °C and 50%, respectively. The working distance varied from 0.4 m to 2.8 m in 0.2 m intervals. While the IEC 80601-2-59 standard suggests a minimum coverage of 20 × 20 pixels for ETRS, at the 2.8 m working distance for IRT-2, only 15 × 15 pixels were usable for ETRS temperature calculation. The same numbers of pixels were used to calculate ETRS and CS temperature ($T_{IRT}$ and/or $T_{ST}$). Additionally, two $T_{ETRS}$ values (35 °C and 37 °C) were selected to assess the effect of $T_{ETRS}$ on measurement accuracy at different working distances.

### 3.5. Computer Simulations

Computer simulations were conducted to assess the effects of ambient temperature, RH, working distance, and the utilization of an ETRS, on IRT and ST accuracies. A wide range of these factors were simulated, including ambient temperature ranging from 15 °C to 35 °C, RH ranging from 5% to 95%, and working distance ranging from 0.2 m to 3.0 m (as indicated in the last line of Table 3). These ranges were intentionally chosen to exceed the recommendations specified in standards, enabling a comprehensive analysis of the effects of these factors on temperature measurements.

We employed MATLAB software (MathWorks) to perform the simulations using the equations in Section 2 and Section 3.4. During the simulations, *τ* was computed using Equations (7) and (8), and the total radiation ($E_{e,total}$) received by the IRTs was determined using Equation (5) based on the calculated *τ*. An object (referred to as CS in the experiments) temperature of 37 °C was assumed. Using $E_{e,total}$, the object temperature was calculated with Equation (6), assuming *τ* is equal to one (i.e., ignoring the effect of ambient temperature, RH, and working distance on *τ* for temperature calculation). The rationale behind this was that the actual energy received by an IRT sensor is dependent on *τ*. When calculating the object temperature using the received energy, errors can occur if environmental factors are not considered (i.e., assuming *τ* = 1). The calculated temperatures were then compared with the actual object temperature (referred to as $T_{CS}$ in the experiments) to estimate the errors and demonstrate the effects of ambient temperature, RH, and working distance under two conditions: without an ETRS and with an ETRS set at 35 °C. The $T_{ETRS}$ can also affect accuracy. To evaluate this effect, the object temperature was set at 37 °C and $T_{ETRS}$ was varied from 20 °C to 50 °C during the simulations. Furthermore, to compare simulation results with benchtop measurement results, we systematically varied one parameter at a time while keeping all other variables constant during simulations, mirroring the methodology detailed in Section 3.4.

## 4. Results

### 4.1. Effect of ETRS Set Temperature

Figure 3 demonstrates the accuracy of IRT readout for a $T_{CS}$ range of 30 °C to 40 °C, considering different $T_{ETRS}$ values within the same temperature range. The ETRS compensation correction was not applied in the data demonstrated in Figure 3, and the legend is shown only for informative purposes. Therefore, different curves in each figure show the stability of each IRT and can be considered as their repeatability performance.

Figure 4 demonstrates the effect of $T_{ETRS}$ on ST accuracy within the temperature range of 30 °C to 40 °C, while maintaining $T_{CS}$ in the same range. This figure can be considered as Figure 3 after ETRS temperature compensation. The horizontal dashed lines in the figure represent the recommended laboratory accuracy between −0.5 °C and 0.5 °C as per the IEC 80601-2-59 standard [32], which include the combined standard uncertainty “u” (including drift, stability, uniformity, etc.) and offset errors ($T_{ST}-T_{CS}$). In a previous study (refer to Table 4 in [33]), we determined |u| for ST-1 and ST-2 to be 0.13 °C and 0.11 °C, respectively. The recommended maximum offset errors $\pm \left|T_{ST}-T_{CS}\right|$ can be calculated as ±|0.5−|u||, as indicated by the horizontal solid lines in the figure. Furthermore, the rectangular gray area denotes the required evaluation range of 34 °C to 39 °C. In our previous clinical study (refer to Table 5 in [10]), we identified the optimal cut-off temperatures for EBT detection, based on maximum facial temperature, to be 36.29 °C and 36.87 °C for ST-1 and ST-2, respectively, with an oral threshold temperature of 38 °C. Considering these findings, we performed our measurements with the ETRS set at 37 °C, aligning with the established cut-off values for accurate and reliable temperature assessments in detecting EBT.

### 4.2. Effect of Viewing Angle

Figure 5 illustrates the difference between $T_{CS}$ and its values measured by both STs ($T_{ST}$). The data analysis revealed that the measurement error increases with viewing angle. For viewing angles below 30°, the measurement error remained below 0.05 °C. For viewing angles ranging from 30° to 40°, the temperature differences remained below 0.1 °C. However, the errors became more apparent for angles exceeding 40°. Notably, at a viewing angle of 60°, the temperature underestimation was measured to be 0.46 °C for ST-1 and 0.35 °C for ST-2. Moreover, at a viewing angle of 75°, the temperature underestimation was found to be 1.21 °C for ST-1 and 1.34 °C for ST-2. These results emphasize the sensitivity of both STs to varying viewing angles and the corresponding decrease in measurement accuracy as the viewing angle increases.

### 4.3. Effects of Ambient Temperature, RH, and Working Distance

A battery of benchtop measurements was performed to investigate the effects of the ambient temperature (ranging from 18 °C to 32 °C), RH (ranging from 15% to 80%), and working distance (ranging from 0.4 m to 2.8 m) on the accuracy of IRTs and STs. $T_{CS}$ was maintained at 37 °C for these measurements. Figure 6 demonstrates the results under various ambient RH and temperatures. The offset errors range from −0.65 °C to 0.65 °C, −0.97 °C to −0.08 °C, −0.12 °C to 0.10 °C, and −0.10 °C to 0.05 °C for IRT-1, IRT-2, ST-1, and ST-2, respectively, within the ambient RH range of 15% to 80% (Figure 6a). Similarly, the offset errors range from −0.30 °C to 0.64 °C, −0.75 °C to −0.14 °C, −0.11 °C to 0.04 °C, and −0.06 °C to 0.02 °C for IRT-1, IRT-2, ST-1, and ST-2, respectively, within the ambient temperature range of 18 °C to 32 °C (Figure 6b). It was observed that the ambient RH and temperature within the investigated ranges had no significant effects on the accuracy of any of the IRTs and STs.

Figure 7 demonstrates the effect of working distance ranging from 0.4 m to 2.8 m with 0.2 m increments on the accuracy of IRTs and STs, while maintaining the ambient temperature at 24 °C and RH at 50%. For $T_{ETRS}$ at 37 °C, the offset errors range from 0.64 °C to 0.76 °C, −0.79 °C to −0.36 °C, −0.09 °C to 0.07 °C, and −0.09 °C to 0.07 °C for IRT-1, IRT-2, ST-1, and ST-2, respectively, within the working distance range of 0.4 m to 2.8 m (Figure 7a). For $T_{ETRS}$ at 35 °C, the offset errors range from 0.59 °C to 0.86 °C, −0.75 °C to −0.38 °C, −0.22 °C to 0.04 °C, and −0.14 °C to 0.01 °C for IRT-1, IRT-2, ST-1, and ST-2, respectively, within the working distance range of 0.4 m to 2.8 m (Figure 7b).

### 4.4. Computer Simulations to Investigate Effects of Ambient Temperature, RH, and Working Distance

Computer simulations were performed to demonstrate the effects of the ambient temperature, RH, and working distance on the measurement accuracy of an IRT or ST, considering an object temperature of 37 °C. $T_{ST}$ was mathematically calculated by assuming $T_{ETRS}$ at 35 °C.

Figure 8a illustrates the estimated *τ* considering the external factors, while Figure 8b demonstrates the $E_{e,total}$ estimation received by an IRT device. Figure 8c,d demonstrate the temperature estimation of a 37 °C object without ETRS and with $T_{ETRS}$ at 35 °C, respectively. These results demonstrate a maximum error of approximately 0.29 °C for an IRT, which can be reduced to 0.04 °C for an ST by utilizing the ETRS temperature compensation.

Figure 9 demonstrates the effect of $T_{ETRS}$ on temperature estimation with $T_{CS}$ at 37 °C. For this simulation, we considered the influence of the external factors on calculation of the $E_{e,total}$ received by an IRT (Equation (5)). To calculate the temperatures of the object and ETRS based on $E_{e,total}$, *τ* was assumed to be one to simulate errors caused by inaccurate *τ*. The calculated $T_{ETRS}$ was used to compensate the calculated object temperature to increase accuracy. The results show that the temperature estimation error can vary from −0.33 °C to 0.22 °C for $T_{ETRS}$ of 20 °C and 50 °C, respectively. However, the error is minimal, approximately ±0.02 °C, for $T_{ETRS}$ of 36 °C and 38 °C.

Figure 10 presents simplified two-dimensional curves, extrapolated from the intricate three-dimensional simulation results. These curves visually depict the simulation outcomes concerning the impact of ambient RH, ambient temperature, and distance on IRT and ST measurement accuracies, all simulated under a specific condition where both $T_{CS}$ and $T_{ETRS}$ were set at 37 °C. From Equations (7) and (8), it is evident that all these parameters influence the *τ* value. If we were to neglect these parameters and assume *τ* = 1, the directly calculated temperatures would exhibit errors in the absence of an ETRS for temperature compensation.

In all three graphs within Figure 10, ST errors were negligible due to temperature compensation based on $T_{ETRS}$. However, the IRT errors, which stem from the simulations disregarding the infrared radiation absorbed by the atmosphere and assuming that the detected radiation comprises all the radiation emitted by the subject (i.e., assuming *τ* = 1), ranged from −0.04 °C to −0.09 °C, −0.05 °C to −0.08 °C, and −0.05 °C to −0.14 °C across varying ambient RH (15% to 80%), ambient temperature (18 °C to 32 °C), and working distance (0.4 m to 2.8 m) ranges, respectively.

## 5. Discussion

In our previous studies, we investigated the essential requirements for IRTs/STs, including stability, drift, uniformity, minimum resolvable temperature difference, and laboratory accuracy and demonstrated the effectiveness of the IRT systems in both benchtop measurements and clinical studies [10,14,33]. External factors (e.g., viewing angle, $T_{ETRS}$, ambient temperature, RH, working distance) in these studies were well controlled and their effects on temperature measurement accuracy remained unexplored. The purpose of this study was to delve into these previously unexamined factors and investigate their effects on laboratory benchtop measurements.

### 5.1. Effect of ETRS Set Temperature

Figure 3 demonstrates the accuracy of the IRTs at different $T_{CS}$ without using the ETRS. The results show higher stability for IRT-1 compared to IRT-2, which agrees with our previous study (see Figure 2 of our previous study [33] for more details). The stability of an IRT can be improved by using a stable ETRS to form an ST. However, $T_{ETRS}$ can affect the ST accuracy. Figure 4 illustrates the importance of $T_{ETRS}$ on the accuracy of the STs. From this figure, the highest accuracy was achieved for $T_{ETRS}$ within the range of 36 °C to 37 °C, which agrees with the cut-off temperature in our clinical study (refer to Table 5 in [10]), specifically for ST-1. While Figure 3 suggested that IRT-2 exhibited higher instability compared to IRT-1, Figure 4 demonstrated that the use of a stable ETRS in ST-2 resulted in significantly improved stability compared to ST-1. When $T_{ETRS}$ was within the range of 36 °C to 37 °C, the offset errors ($T_{ST}-T_{CS}$) for both IRTs met the accuracy requirements (i.e., within the two horizontal solid lines in Figure 4), except for when $T_{CS}$ was 34 °C and $T_{ETRS}$ was 37 °C and when $T_{CS}$ was 39 °C and $T_{ETRS}$ was 36 °C for ST-1.

### 5.2. Effect of Viewing Angle

There is a correlation between the skin temperature measurements and the viewing angles [67,68]. Cheng et al. showed that temperature underestimation becomes significant when viewing angles were beyond 40° on flat surfaces [36]. Muniz et al. demonstrate errors exceeding 2% at viewing angles above 35° with an IRT featuring an 18° and 35° field of view (FOV), and above 25° with an IRT having a 7° FOV [68].

Previous investigations on the effect of viewing angle were limited to short-wavelength IRTs or long-distance measurements in a non-medical setting (e.g., infrastructure, building, and satellite imaging). In our study, we designed the parameters (e.g., IRT wavelength, working distance) to mimic the application of EBT screening. We determined that viewing angles below 30° achieved an error below 0.05 °C, while wider viewing angles exceeding 40° showed significant underestimations of 0.1 °C up to 1.3 °C for an IRT with FOV larger than 14° (Figure 5). For an IRT with a smaller FOV than 14°, a viewing angle of less than 25° might be preferred [68]. These results highlight the importance of properly aligning viewing angles to minimize measurement artifacts and ensure accurate temperature measurements, in agreement with studies conducted for other applications [36,38,69]. Other factors that might change the effects of the viewing angle include the wavelength region of the IR sensor, instantaneous field of view, object material, and object curvature [36,38,40,68,69].

### 5.3. Effects of Ambient Temperature, RH, and Working Distance

Our previous clinical studies have demonstrated the high sensitivity and specificity of IRTs in detecting EBT when controlling the ambient room temperature and RH [10,14]. Table 3 presents a comparison of the ambient temperature, ambient RH, and working distance ranges for various studies. In this study, we investigated the effects of these factors, considering that certain conditions may necessitate broader room temperature and RH ranges than the current standard recommendations. For instance, maintaining a high RH (e.g., ~70%) and ambient temperature (e.g., ~32 °C) is outside of standard recommendations but can be beneficial for premature neonates [70]. Figure 6 shows the effects of ambient temperature and RH. As anticipated, the STs exhibit significantly superior accuracy compared to the IRTs, strongly indicating that the inclusion of a stable ETRS can greatly enhance accuracy across the wide ranges of ambient temperature and RH examined in this study. Particularly noteworthy is the improved accuracy achieved through the proper use of the ETRS in an optimal set temperature range of 36 °C to 37 °C, which helps minimize the impact of environmental factors. These results indicate that the ambient temperature and RH within the examined ranges had minimal impact on the accuracy of temperature measurements with an ST in the benchtop setting.

Since our study relied on bench tests without human subjects, we should be careful to extend the conclusions regarding the effects of ambient temperature and RH to clinical measurements. Human thermal sensation is primarily influenced by the overall thermal balance of the body, which is affected by various factors including physical activities, clothing levels, and environmental parameters, such as air temperature, mean radiant temperature, air velocity, and air humidity [71]. Conducting a systematic study encompassing all these factors is challenging, and there is no consensus on the proper ranges of these factors to define thermal comfort. However, assuming that other factors are within their optimal ranges or conditions, we will only focus our discussion on ambient temperature and RH. The neutral temperature, often referred to as the comfort temperature, is typically defined as the environmental temperature range within which the oxygen consumption, and thus heat production of homoiothermic subjects, is minimal. For an un-clothed adult, this range falls between 26 °C and 31 °C [72]. Ahmed’s study indicated that under still air conditions and an average RH of 70%, the boundaries of average air temperature for outdoor comfort can vary between 28.5 °C and 32.8 °C [73]. Jing et al. found that about 80% of subjects could not tolerate a thermal environment of 30 °C and 80% RH [74]. It appears that humans are less sensitive to RH compared to ambient temperature. Fountain et al. did not observe a clear difference in humidity response among sedentary subjects within the ranges of 20 °C/60% RH to 26 °C/90% RH [75]. Generally, a 10% increase in RH and a 0.3 °C rise in ambient temperature are both perceived as contributing equally to a warmer sensation [71]. Furthermore, Djamila et al. found that the occupants were thermally comfortable within a wide range of RH, with the mean RH, corresponding to a neutral temperature of 30 °C, being about 73% [76]. The ASTM E1965-98 standard specifies an operating temperature range of 16 °C to 40 °C and an RH range of up to 95% for NCIT applications [42]. Considering that IRTs and NCITs share similar physical principles, these ranges can be used as references. Taking all of these studies, including our own [14], into consideration, it is reasonable to extend the current ambient temperature range (20 °C to 24 °C) and RH range (10% to 50%) to encompass range of 20 °C to 29 °C and 10% to 80% RH, assuming the subject’s skin is dry. Moist skin surfaces resulting from sweating might affect skin emissivity [44] and lead to higher evaporation rates, which in turn can affect surface temperature measurements [77].

Figure 7 shows the effect of the working distance on the benchtop measurements for two sceneries of $T_{ETRS}$ = 37 °C and $T_{ETRS}$ = 35 °C, ranging from 0.4 m to 2.8 m. Distances closer than 0.4 m were not studied because capturing images of both CS and ETRS simultaneously became challenging. The standard recommends that the face should be covered by a minimum of 180 × 240 pixels, while the ETRS region should be covered by at least 20 × 20 pixels [32]. However, it is important to note that the actual number of pixels covering the subject’s face may vary depending on the working distance between the camera and the subject. In our previous studies, a working distance of 0.6 to 0.8 m was adopted to meet the standard criteria for a minimum of 180 × 240 pixels to cover the individual’s face [10,14,33]. IRTs with different pixel counts and FOV specifications require varying working distances to align with standard recommendations. Based on Section 4.1, the optimal ETRS set temperature for EBT screening is between 36 °C and 37 °C. To evaluate the results beyond this optimal range, Figure 7 demonstrates the difference when $T_{CS}$ was kept at 37 °C and 35 °C, respectively. The results indicate that when $T_{ETRS}$ was 37 °C, the STs exhibited smaller errors compared to when $T_{ETRS}$ was 35 °C. The working distance has little effect on IRT and ST accuracies in the working distance range we studied.

### 5.4. Computer Simulations to Investigate Effects of Ambient Temperature, RH, and Working Distance

The simulation results in Figure 8 illustrate the effects of environmental factors and working distance on *τ*. The simulation results encompass the $E_{e,total}$ received by an IRT, considering the actual *τ* and the temperature readout under the assumption of *τ* equal to one. These results are presented for both scenarios, with and without the inclusion of an ETRS. Figure 9 illustrates the effect of different $T_{ETRS}$ values on the accuracy of temperature readouts. The results in Figure 8 and Figure 9 highlight the significance of using an ETRS to minimize the effects of external factors on temperature measurements. It is observed that optimal accuracy was achieved when $T_{ETRS}$ was close to the target temperature or threshold temperature. These simulations show how temperature measurement accuracy can be influenced by ambient temperature, RH, working distance, and $T_{ETRS}$.

Results in Figure 10 demonstrate the influence of ambient RH, ambient temperature, and working distance, with *τ* set to one to ignore these parameters, while keeping all other parameters constant, mirroring the benchtop results depicted in Figure 6 and Figure 7a. A meaningful comparison of Figure 10a with Figure 6a, Figure 10b with Figure 6b, and Figure 10c with Figure 7a can yield valuable insights since the default parameter values are the same in these figures. The simulation results in Figure 10 reveal that ambient RH, ambient temperature, and measuring distance can directly impact *τ*, thus affecting the total radiation received by the IRT sensor and the resulting IRT temperature readings. The absolute errors introduced by changes in ambient RH, temperature, and distance are less than 0.14 °C across the simulated parameter ranges. In the experimental data presented in Figure 6 and Figure 7a, proper *τ* values were utilized based on ambient RH, ambient temperature, and measuring distance, effectively mitigating the influence of these parameters in theory. This explains the absence of discernible trends in experimental errors as these parameters gradually vary. In Figure 6 and Figure 7a, experimental errors for STs range from −0.12 °C to 0.10 °C, and for IRTs, they span from −0.97 °C to 0.76 °C, encompassing all the parameter ranges under study. The experimental errors for IRTs are notably larger than the simulated errors. The primary reason behind this disparity is that the simulations assume an ideal infrared sensor, while inaccuracy in a real sensor typically has a significant impact on measurement errors.

### 5.5. Study Limitations

In this study, the measurements were conducted indoors without any external sources of infrared radiation. Therefore, the ambient/atmosphere temperature and reflected temperature were assumed to be equal. The American Society for Testing and Materials (ASTM) has provided a test method for measuring the reflected temperature [56], which can be another factor to consider in accurate temperature measurements.

The effect of wind or airflow, which can affect EBT screening accuracy, was not discussed in this paper. However, the ISO standard suggests that the screening area should be free from significant natural and forced convective airflow to ensure accurate measurements [30].

We recommend maintaining an optimal viewing angle below 30° when assuming a flat skin surface and a FOV larger than 14°. However, it is important to consider that the skin has varying levels of curvature at different locations. Depending on the specific area from which thermal radiation will be captured for temperature calculation, these variations in curvature should be considered.

Our study primarily focuses on assessing the influence of ambient temperature and RH on infrared signal detection. Moreover, it is important to note that these two factors, along with others, can affect human thermoregulation and thermal comfort [43]. Consequently, they can influence the relationship between the temperature measured at the measurement site and the reference site temperature [14]. Other factors include physical activities (sedentary and non-sedentary), clothing levels, mean radiant temperature, and air velocity [71,75]. The sweating threshold can vary among individuals and can be affected by various factors, such as subject’s fitness level, environmental conditions, adaptation period, and the RH [78,79]. Changes in RH have a direct relationship with the sweating rate [80], and gender difference can affect sweating capacity (with females generally having lower sweating capacity than males) [81]. Additionally, people in different regions may have diverse perceptions of comfort [82,83], and body mass index can potentially affect temperature measurements, particularly in relation to skin temperature [84]. The investigation of environmental factors that can affect biological/physiological changes was not within the scope of this study. The objective of this study was limited to the benchtop evaluation of the effect of ambient temperature and RH on infrared signal detection, although we did discuss their possible range for clinical measurements.

It is worth mentioning that different models exist for estimating τ [46,55,57] besides the one we used. Minkina et al. presented various models for evaluating τ for long-wavelength (7–14 µm) and short/mid-wavelength (2–5 µm) IR radiation [46]. We did not compare different models for estimation of τ. The model employed in this study to estimate *τ* is based on the equation provided by the manufacturer of IRT-1 [48].

Our primary aim is to objectively and quantitatively assess the effects of environmental and deployment factors on IRT accuracy and quantify their proper ranges through benchtop measurements and computer simulations. It is important to acknowledge that an accurate temperature measurement can be influenced by a multitude of other factors. Skin temperature naturally fluctuates due to variations in cardiac rhythm throughout the day [17,85,86], and the magnitude of these fluctuations can be influenced by gender and the specific measurement location [85,86]. Physical activity can increase heat production in muscles, impacting body surface temperature, which is why it is advisable to avoid physical activity before temperature measurement [87]. While IRT holds potential for targeted exercise scenarios and sports medicine [87,88,89], our study does not cover these areas. The impact of skin color has been extensively studied in the context of pulse oximeters. However, there is no strong evidence to suggest that skin pigmentation significantly affects IRT accuracy [64]. This is because pulse oximeters gauge the amount of light passing through the skin, where pigmentation can be a factor for light absorption. In contrast, IRTs measure the infrared light emitted from the skin due to the vibration of water and organic molecules, a proceed that is similar across all skin colors.

Several documents offer valuable guidance on best practices for measuring EBT with IRTs. The ISO/TR 13154 standard document [30] outlines guidelines for the deployment, implementation, and operation of IRTs for EBT screening. Additionally, best practice guide [31] offers detailed recommendations and operating procedures for conducting human body temperature screening using IRTs. These documents encompass best practices related to acclimation time, measuring background, sources of radiant heat, obstructions, skin conditions, etc. Due to space constraints, we are unable to comprehensively discuss all the factors that can potentially affect IRT accuracy in this study.

Finally, our study exclusively addresses the detection of EBT using facial thermal images. The necessary minimum resolution is determined with this specific focus. Therefore, we should exercise caution when attempting to extrapolate these results to other medical applications. Additionally, it is important to note that the minimum resolution is defined by pixel count. However, having more pixels in an IRT does not necessarily equate to superior performance compared to the one with fewer pixels. Other parameters, such as instantaneous field of view [90], stability and drift, image uniformity [33], modulation transfer function [91], signal transfer function, noise [92], among others, can have a significant impact on the performance of an IRT.

## 6. Conclusions

We conducted a comprehensive study combining benchtop measurements and computer simulations to examine the impact of environmental factors (ambient temperature and RH) and deployment variables ($T_{ETRS}$, working distance, and object orientation/viewing angle) on the accuracy of IRTs and STs for measuring body temperature. Results emphasized the importance of appropriate calibration and environmental control for reliable temperature readings. Comparison between simulation and experimental data also shows that the hardware performance might be more important than the environmental and deployment parameters for accurate thermal radiation detection. On the other hand, environmental parameters may exert a more significant influence on human body thermal regulation than on thermal signal detection, a topic not covered in this paper.

We have provided quantitative definitions of acceptable ranges for the environmental and deployment parameters under investigation. Some of these ranges have been expanded to accommodate IRT applications in less-than-ideal scenarios. The insights gained from our research inform the development of streamlined test methods and support ongoing efforts in establishing IRT guidelines. Our results will refine IRT standards and promote their practical use in healthcare settings, enhancing the ability to swiftly identify and address potential health risks.

## Figures and Tables

**Figure 1 sensors-23-08011-f001:**
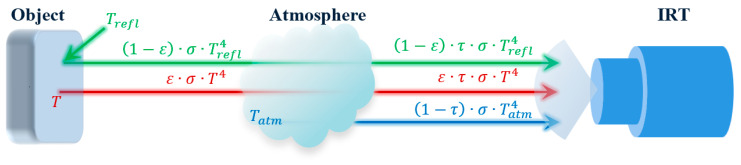
Principle of the total radiation received by an IRT. [σ: Stefan–Boltzmann constant, ε: emissivity, τ: atmospheric transmittance, $T_{refl}$: reflected temperature, *T*: object temperature, $T_{atm}$: atmosphere temperature].

**Figure 2 sensors-23-08011-f002:**
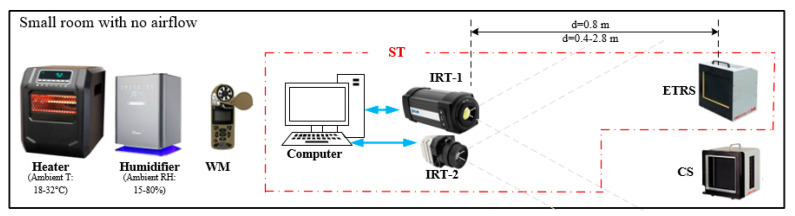
Schematic of the experimental setup. The red dash-dotted box shows the ST systems.

**Figure 3 sensors-23-08011-f003:**
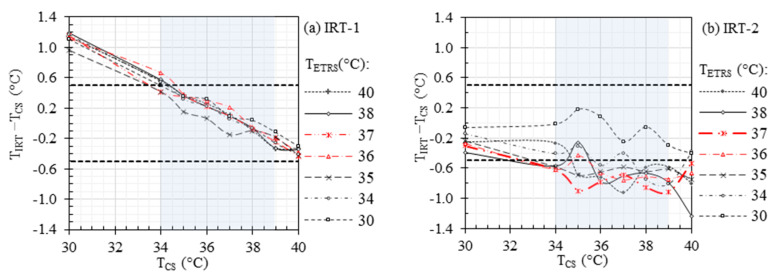
Accuracy of (**a**) IRT-1 and (**b**) IRT-2 without ETRS compensation. Horizontal dashed lines represent the recommended laboratory accuracy and the rectangular gray area denotes the required evaluation range of 34 °C to 39 °C.

**Figure 4 sensors-23-08011-f004:**
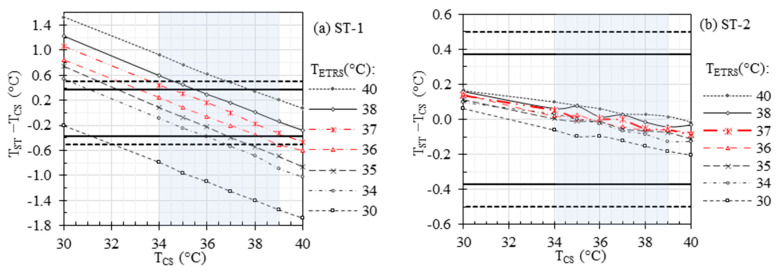
Effects of temperature compensation and $T_{ETRS}$ on the accuracy of (**a**) ST-1 and (**b**) ST-2. Horizontal dashed and solid lines represent the recommended laboratory accuracy and offset errors, respectively, and the rectangular gray area denotes the required evaluation range of 34 °C to 39 °C.

**Figure 5 sensors-23-08011-f005:**
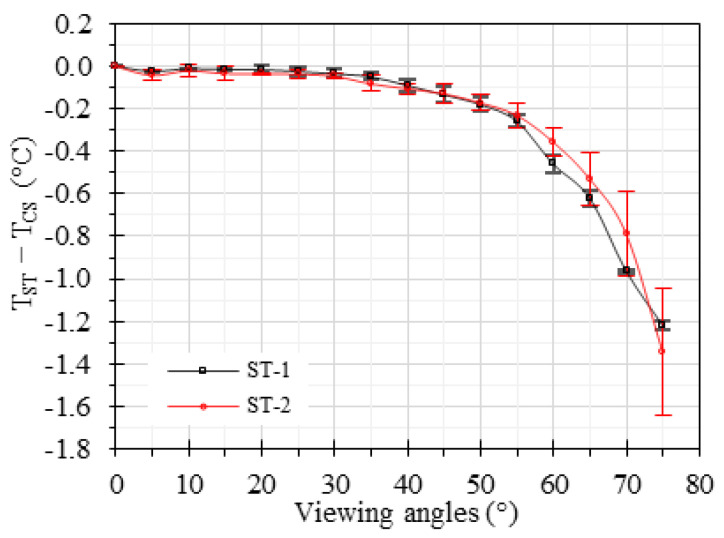
Effect of viewing angle on temperature accuracy for the two STs. The error bars represent the standard deviation.

**Figure 6 sensors-23-08011-f006:**
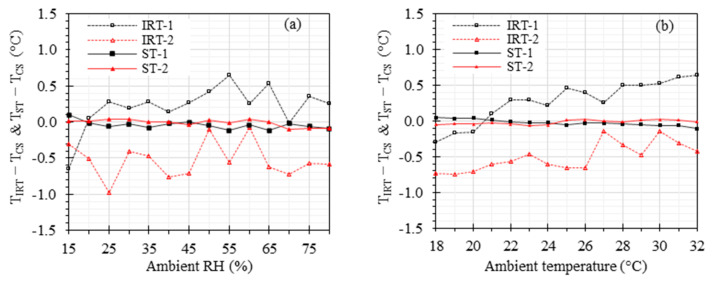
Effects of ambient RH and temperature: $T_{IRT}-T_{CS}$ and $T_{ST}-T_{CS}$ versus (**a**) ambient RH with ambient temperature at 24 °C and (**b**) ambient temperature with ambient RH at 35%. The working distance was kept at 0.8 m.

**Figure 7 sensors-23-08011-f007:**
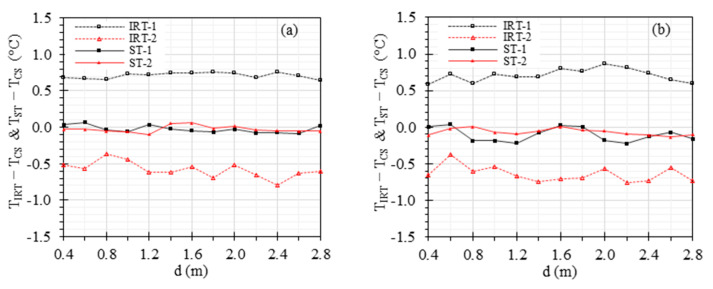
Effect of the working distance: $T_{IRT}-T_{CS}$ and $T_{ST}-T_{CS}$ versus working distance with (**a**) $T_{ETRS}$ = 37 °C and (**b**) $T_{ETRS}$ = 35 °C.

**Figure 8 sensors-23-08011-f008:**
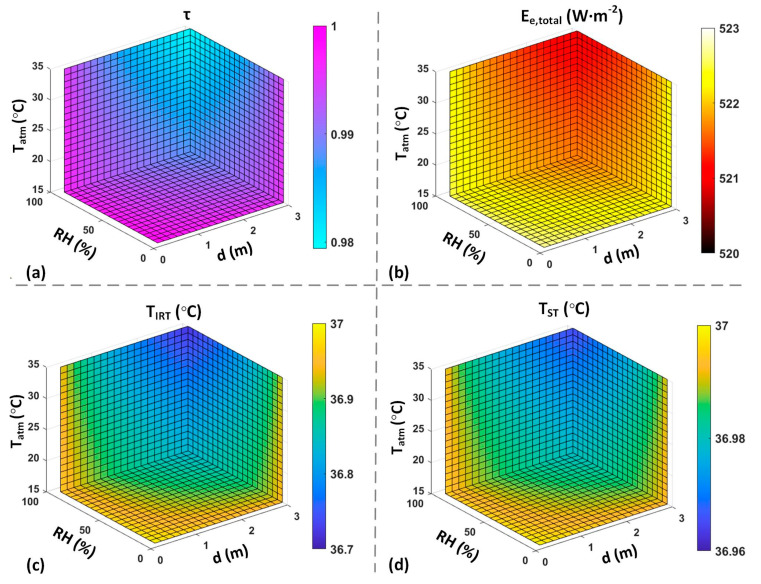
Computer simulation results with $T_{CS}$ = 37 °C. ((**a**): estimated *τ* based on environmental factors. (**b**): $E_{e,total}$ received by IRT based on the estimated *τ*. (**c**): calculated $T_{CS}$ measured by an IRT assuming *τ* = 1. (**d**): calculated $T_{CS}$ measured by an ST assuming *τ* = 1 and $T_{ETRS}$ = 35 °C).

**Figure 9 sensors-23-08011-f009:**
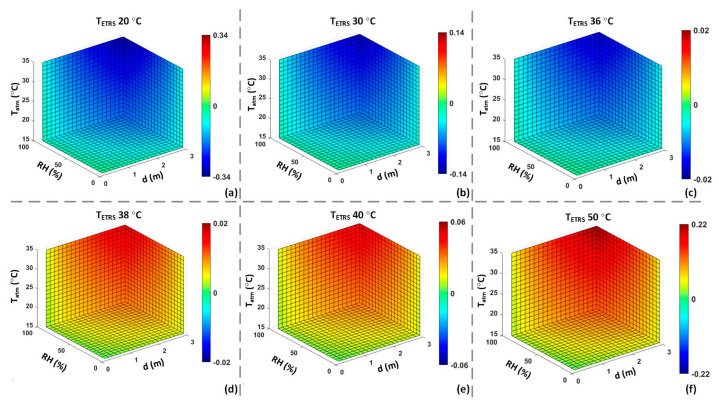
Computer simulation of $T_{ETRS}$ effect: Offset error $\left(T_{ST}-T_{CS}\right)$ of an ST to measure a CS at 37 °C with different $T_{ETRS}$ values assuming *τ* = 1. ((**a**): $T_{ETRS}$ = 20 °C; (**b**): $T_{ETRS}$ = 30 °C; (**c**): $T_{ETRS}$ = 36 °C; (**d**): $T_{ETRS}$ = 38 °C; (**e**): $T_{ETRS}$ = 40 °C; (**f**): $T_{ETRS}$ = 50 °C).

**Figure 10 sensors-23-08011-f010:**
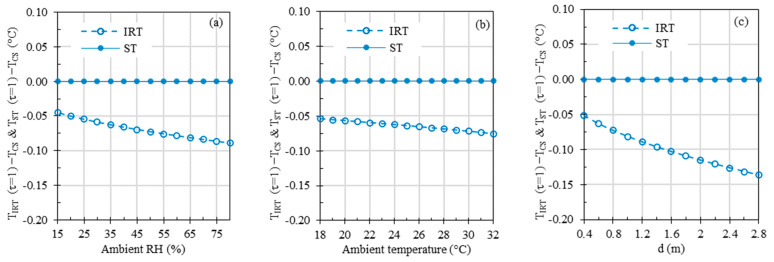
Computer simulations depicting the effects of (**a**) ambient RH, (**b**) ambient temperature, and (**c**) working distance, with both $T_{CS}$ and $T_{ETRS}$ set at 37 °C. If not specified, the default values for ambient temperature and distance were set to 24 °C and 0.8 m, respectively. For the sake of comparison, the ambient RH values were set at 35% and 50% in cases (**b**,**c**), mirroring the experimental conditions depicted in Figure 6b and Figure 7a.

**Table 1 sensors-23-08011-t001:** Equipment list.

Device Names	Models and Manufacturers	Abbreviation	Functions and Specifications [33]
Infrared thermographs (IRTs)	A325sc, FLIR Systems Inc., Nashua, NH, USA	IRT-1	Measure test target temperature. Part of an ST, 320 × 240 pixels, spectral range of 7.5–13 µm, Field of view (FOV) 17° and 14° [horizontal and vertical].
	8640 P-series, Infrared Cameras Inc., Beaumont, TX, USA	IRT-2	Measure test target temperature. Part of an ST, 640 × 512 pixels, spectral range of 7–14 µm, FOV 30° and 25° [horizontal and vertical].
Extended area blackbodies	SR-33N-4, CI Systems Inc., Simi Valley, CA, USA	ETRS	Work as an external temperature reference source for offset compensation. Part of an ST.
	SR-800R-4D, CI Systems Inc., CA	CS	Serve as a calibration source or test target.
Humidifier	EE-6913, Crane-USA, Itasca, IL, USA	-	Control RH in the range of 15–80%.
Heater	HT1188, Supply Chain Sources LLC, Brea, CA, USA	-	Control $T_{atm}$ in the range of 18–32 °C.
Weather meter	Kestrel 4500NV, Weather Republic LLC, Downingtown, PA, USA	WM	Measure $T_{atm}$ and RH, which can be used to assess *τ*.

Note: In our study, we refer to the graybodies ETRS and CS as blackbodies to align with their commonly used names, considering their high emissivity.

**Table 2 sensors-23-08011-t002:** IRT-1 camera parameters [47,48,57].

$\alpha_{1}$	$\alpha_{2}$	$\beta_{1}$	$\beta_{2}$	$h_{1}$	$h_{2}$	$h_{3}$	$h_{4}$	$K_{atm}$
0.006569	0.01262	−0.002276	−0.00667	1.5587	6.939 × 10^−2^	−2.7816 × 10^−4^	6.8455 × 10^−7^	1.9

**Table 3 sensors-23-08011-t003:** Environmental factor ranges.

	Ambient Temperature (°C)	Ambient RH (%)	Working Distance (m)
ISO/TR 13154 [30]	20–24	10–50	NA
IEC 80601-2-59 [32] (Clause 201.7.9.3.1)	<24	<50	NA
Zhou et al. [10]	20–24	10–62	0.6–0.8
Wang et al. [14]	20–29	10–62	0.6–0.8
Martinez-Jimenez et al. [62]	23	40	0.3
Ng et al. [63]	20–25	40–75	NA
Charlton et al. [64]	28–30	19–30	NA
Ring et al. [65]	20–21	NA	<1
Healy et al. [66]	19.8–22.6	53–70	0.5–0.8
Current Benchtop	18–32	15–80	0.4–2.8
Current Simulations	15–35	5–95	0.2–3.0

## Data Availability

Not applicable.

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
