# Peer review of "Best Practices for Body Temperature Measurement with Infrared Thermography: External Factors Affecting Accuracy"

_sensors, 2023, doi:10.3390/s23188011_

Round 1
Reviewer 1 Report
Dear Authors,
The work is interesting, however I see some flaws, mentioned below:
Introduction:
You have neglected to reference essential publications that address the issue of standardization procedures in infrared examination with humans. These publications serve as the foundation for the current protocols. To enhance the introduction, I recommend citing the following references:
1. Moreira, D.G., et al.: "Thermographic imaging in sports and exercise medicine: a Delphi study and consensus statement on the measurement of human skin temperature." J. Thermal Biol. 69, 155–162 (2017).
2. Chudecka, M., Lubkowska, A.: "Temperature changes of selected body’s surfaces of handball players in the course of training estimated by thermovision, and the study of the impact of physiological and morphological factors on the skin temperature." J. Thermal Biol. 35(8), 379–385 (2010).
3. Zagrodny, B.: "Standardisation Procedure of Infra-red Imaging in Biomechanics." In: Hadamus, A., Piszczatowski, S., Syczewska, M., Błażkiewicz, M. (eds) Biomechanics in Medicine, Sport and Biology. BIOMECHANICS 2021. Springer, Cham. https://doi.org/10.1007/978-3-030-86297-8_13.
Additionally, You should consider elaborating on the infrared (IR) protocols in the introduction and discussing them more extensively in the later discussion.
Discussion:
The discussion section falls short as it lacks engagement with the findings and results of other authors who explored similar problems. Expanding this section to incorporate comparisons and contrasts with other research in the field would provide a broader context and strengthen the paper's contribution.
Inclusion of Experimental Data:
While the paper is primarily theoretical and simulation-based, where feasible, the results should be supplemented with experimental data. Taking into account other researchers' results and incorporating existing experimental data would lend additional credibility to the findings.
Clearer Comparisons in Simulations and Experiments:
I advise providing more explicit comparisons in sections where simulations and experiments are presented. Clear and concise descriptions of how the results of the simulations and experiments correlate would enhance the reader's understanding.
Addressing these flaws will not only enhance the quality and comprehensiveness of the paper but also contribute to the existing knowledge in the field of infrared scanning of the human body.
Reviewer 2 Report
The paper is well written. It is a nice overview of different effects on IRT measurement. However, it doesn't bring much, if any, new information.
Here are some additional notes:
I suggest the replacement of the word thermographs in the title for thermography. This way you may reach a broader audience.
Chapter 2 seems pointless. Anyone working with IR cameras should know this. Consider leaving It out.
Line 212 working distance of d=0.8m to satisfy this requirement, as discussed in our previous publication [16]. I believe it would be beneficial for reader to state why and then more information can be found in our previous publication [16]
What tool did you use for simulations? I could not find this information.
Line 528 there is a typo: 32 8 , it should be 32.8
In the discussion chapter, it is not explained why that effect occurs. E.g., why there is such an effect of viewing angles? Why ETRS should be 36-37?
Reviewer 3 Report
Thanks to the authors for their vast investigation and submitted manuscript.
The manuscript addresses a general problem in IR Thermography with a specific focus on the healthcare application. This can be of interest to many individuals and organizations to broaden their view and optimize their approaches.
However, despite the good experimental approach, well-presented results and the conclusions provided, this manuscript only addresses a very standard approach that can be found in any textbook related to IR Thermography and guidelines provided in the reference standards. The theories presented can be found in any relevant textbook and are not specific to this study. Moreover, as mentioned in section 5.5 by the authors, the main issues that arise in a common practical IR Thermography of a live object under realistic circumstances (mostly moving target, uncontrolled environment, variable skin colours and physical conditions, etc.) haven't been considered in this research. If the main aim of the manuscript is to provide a best practice, it has to address at least a few of these variable parameters that may affect a real-world scenario, which clearly are not the viewing angle, room temperature and relative humidity (as these are the main three obvious test parameters in any IR Thermography practice).
Therefore, I would like to ask the authors to provide their justifications and revise their manuscript to be more attractive to the readers.
Best of luck.
